# An Ideal-Type Analysis of People’s Perspectives on Care Plans Received from the Emergency Department following a Self-Harm or Suicidal Crisis

**DOI:** 10.3390/ijerph20196883

**Published:** 2023-10-04

**Authors:** Sally O’Keeffe, Mimi Suzuki, Rose McCabe

**Affiliations:** 1Population Health Sciences Institute, Newcastle University, Newcastle-Upon-Tyne NE2 4AX, UK; 2Unit for Social and Community Psychiatry, Queen Mary University of London, London E13 8SP, UK; mimi.suzuki@nhs.net; 3School of Health and Psychological Sciences, City, University of London, London EC1R 1UW, UK; rose.mccabe@city.ac.uk

**Keywords:** emergency department, ideal-type analysis, liaison psychiatry, safety planning, self-harm, suicide prevention

## Abstract

People presenting to Emergency Departments (EDs) in a self-harm/suicidal crisis in England receive a psychosocial assessment and care plan. We aimed to construct a typology of peoples’ perspectives on crisis care plans to explore the range of experiences of care plans. Thirty-two semi-structured interviews with people who presented to EDs following a self-harm/suicidal crisis in England were analysed using an ideal-type analysis. Cases were systematically compared to form clusters of cases with similar experiences of care plans. People’s perspectives on care plans fitted into three types: (1) personalised care plans (*n* = 13), consisting of advice or referrals perceived as helpful; (2) generic care plans (*n* = 13), consisting of generic advice that the person already knew about or had already tried; and (3) did not receive a care plan (*n* = 6) for those who reported not receiving a care plan, or who were only provided with emergency contacts. Care planning in the ED following a suicidal/self-harm crisis was perceived as supportive if it provided realistic and personalised advice, based on what had/had not worked previously. However, many people reported not receiving a helpful care plan, as it was ill-fitted to their needs or was not considered sufficient to keep them safe, which may mean that these patients are at increased risk of repeat self-harm.

## 1. Introduction

There are over 700,000 suicides globally each year [1]. There is an urgent need for interventions for people at increased risk of suicide, including those who self-harm, which is one of the leading clinical risk factors for suicide, associated with a 10-fold increase in the odds of suicide [2]. Psychological interventions exist for people at risk of suicide, including dialectical behaviour therapy [3]. However, there are significant barriers to accessing treatment, including limited availability, long waitlists and inclusion criteria that exclude many people [4].

Over 30% of people who die by suicide presented to hospital in the two years prior to their death [5]. Emergency Departments (EDs) are the only healthcare contact for many people at risk of suicide, providing an opportunity for intervention to mitigate against future suicide risk [6]. In the UK, guidelines state people presenting at a hospital during a self-harm crisis will be seen by an age-appropriate liaison psychiatry or mental health professional for a psychosocial assessment [7], which may reduce risk of repeat self-harm [8,9]. Psychosocial assessments consist of assessment of needs and risk, leading to a care plan. A care plan can be defined as a document outlining the needs of the person and the interventions that will support their recovery [10], although the term is often used interchangeably with treatment plan. It has been recognised that care planning should be a personalised collaborative process, whereby goals and actions are jointly agreed for managing a patient’s long-term health condition [11], reflecting the person’s choices, wishes and preferences [12]. A systematic review found 19 studies of personalised care planning that were found to be associated with improvements in care for diabetes, hypertension and asthma; self-efficacy and self-care; and some improvements in psychological well-being [11]. However, these studies were mostly conducted in primary care, involving multiple contacts, so care planning was an ongoing process. This differs from ED contacts, which are typically brief, one-off contacts, and little is known about how such contacts can be optimised to ensure a person-centred approach despite these constraints. Studies show that patients often have limited involvement in decisions about their care plan [12,13] and that care plans are frequently not personalised to the individual [14]. Research has found that patients are often not aware of what is in their care plan or even that they have one [15]. People who have attended the ED after self-harm have reported variable experiences of care plans [16,17,18]. Research shows the importance of how recommendations are given, with patients more readily accepting recommendations when practitioners acknowledge and validate patients’ views and past experiences [19].

Care plans are broad in scope, and components of safety planning—a specific type of intervention for self-harm—may fall within care plans. Safety plans aim to support patients to cope in times of crisis by planning a set of coping strategies and sources of support [20]. While care plans outline the support and interventions available for the person, safety plans tend to break this down further, specifying a hierarchical series of steps that can be followed in times of crisis, including non-professional coping strategies (e.g., things people can do in their environment or people in their social network whom they can contact for support). Systematic reviews show that safety planning is a promising brief intervention for reducing suicidal behaviour [21,22,23] and the feasibility and acceptability of safety planning in EDs has recently been demonstrated in the UK [24]. Theories propose that suicidal behaviour results from perceived burdensome, thwarted belongingness and capability for suicide [25]. The Integrated Motivational–Volitional model suggests that defeat and entrapment are the primary drivers of suicidal ideation, and factors that explain the transition to act on these thoughts include acquired capability, access to lethal means and impulsivity [26]. Safety planning seeks to disrupt the cycle between distress, suicidal thoughts and behaviour by distracting patients from suicidal thoughts; helping to interrupt unproductive cognitive processes; and increasing connectedness, autonomy and competence in managing their distress [27,28].

The aim of the present study was to construct a typology of people’s perspectives on crisis care plans. The ideal-type analysis facilitated the development of distinct categories of experiences of care plans, whilst also allowing for the recognition of the variation between and within categories of experiences of care plans in the context of one-off ED contacts following a self-harm/suicidal crisis.

## 2. Materials and Methods

### 2.1. Design

The secondary analysis was conducted on patient interviews from two studies exploring treatment for self-harm and suicidality in EDs in England: the Relate study [18] and the ASsuRED study [16].

Relate study: Patients presenting to an ED in England in a self-harm/suicidal crisis were recruited into the study. Within two weeks, they were invited to take part in an interview about their experiences of their psychosocial assessment. The topic guide included a question asking what they thought about their treatment/care plan.

ASsuRED study: People with experience of attending the ED for self-harm were recruited through mental health charities, service user groups and social media. They participated in an interview/focus group exploring their experiences of self-harm care in EDs. The topic guide asked about what was helpful and unhelpful for patients presenting to EDs with self-harm. Although they were not specifically asked about their care plans, this spontaneously came up in most of the interviews and focus groups.

All participants had been seen by a mental health professional in the ED. Further details about recruitment were published elsewhere [16,18]. This study reports on people’s experiences of care/safety plans received in the ED.

### 2.2. Data Collection

Participants took part in a semi-structured interview/focus group to explore their experiences of treatment following a self-harm/suicidal crisis in EDs, with a researcher who was independent of the hospital in which participants were treated. Inclusion criteria were age 16 years or above, capacity to consent and experience of presenting to the ED with self-harm/suicidal thoughts. Interview schedules included the exploration of participants’ experiences of ED care plans. Interviews were conducted by researchers and were recorded and transcribed verbatim. 

Relate participants (*n* = 28) were interviewed between September 2018 and April 2019, within two weeks of having attended the ED. ASsuRED participants (*n* = 19) were interviewed between September and December 2019. Participants were excluded from the present study if they did not discuss a care plan (5 Relate participants and 7 ASsuRED participants). The sample for this study comprises 32 participants.

### 2.3. Data Analysis

An ideal-type analysis was used to construct a typology of people’s perspectives on their care plan from the ED. Ideal-type analysis is a multi-case study qualitative method that seeks to systematically describe naturally occurring patterns of human experience by forming categories or types [29,30]. Compared to methods such as thematic analysis, which identifies themes across individuals, an ideal-type analysis allowed us to provide illustrative cases to exemplify different experiences of care planning. Participants were grouped together into clusters based on shared characteristics, making it possible to appreciate individual differences in people’s experiences of care plans. The seven steps of ideal-type analysis were followed, as described by Stapley and colleagues [29,30]:Familiarisation with the dataset by reading the transcripts.Writing case reconstructions, i.e., a written summary for each participant about his/her perspective on his/her care plan.Constructing ideal types by systematically comparing each case reconstruction to form clusters of cases with similar experiences.Identifying illustrative cases (or “optimal cases”) that best depicted each ideal type.Forming ideal-type descriptions to describe the core characteristics of each ideal type.Credibility checks: An independent researcher grouped cases into the ideal types, using the ideal-type descriptions. There was 94% agreement with the classification of cases, reflecting disagreement with the classification of two cases. The researchers discussed these cases to reach a consensus on where these cases best fitted. The typology was refined to clarify the core characteristics of each type.Comparing similarities and differences in cases within and between the ideal types. Illustrative cases were used to compare all other cases in each ideal type in order to explore the ways in which all other cases reflected/deviated from the illustrative case, based on their demographic data, whether they were known to services and the recommendations in their care plans.

### 2.4. Ethical Considerations

Ethical approval was obtained from the London–Central (Ref:17/LO/1234) and London–Surrey Borders (Ref:19/LO/0778) Research Ethics Committees. Participants provided written informed consent. Identifiable information was removed or disguised, and participants were assigned a pseudonym to protect their identity. Participants were debriefed at the end of their interview or focus group to ensure that their participation had not left them feeling distressed. A risk protocol was in place for researchers to follow in the event that they had concerns about the safety of a participant to ensure that appropriate support was in place (e.g., via signposting or contacting a care provider). 

## 3. Results

The sample comprised 32 participants. The average age was 35 years (range: 17–76 years), 75% were female and the majority was White British (81%).

Using ideal-type analysis, three types of experiences of ED care plans were found: a personalised care plan (*n* = 13), a generic care plan (*n* = 13) and no care plan (*n* = 6).

### 3.1. Ideal-Type 1: Personalised Care Plan

Thirteen participants (41%) received a personalised care plan in the ED.

#### 3.1.1. Ideal-Type Description

Thirteen participants perceived their care plan to be personalised and appropriate. It consisted of at least one recommendation that they considered to be helpful—typically something new to them, such as advice about a service or resource that they were not aware of or that they had not previously considered. This meant that they left the hospital with some advice or recommendations tailored to their needs. Typically, personalised care plans focused on professional sources of support, such as referrals to mental health/voluntary sector services (e.g., crisis team/therapy). Recommendations often spanned across multiple types of services (e.g., General Practitioner (GP), talking therapies and crisis lines) and having numerous recommendations may have increased the likelihood of some of these being perceived as useful. Moreover, recommendations went beyond mental health support (e.g., employment or education advice), making a holistic plan that covered a range of areas of the person’s life. As recommendations were perceived as relevant to their problems, people had taken up these suggestions or intended to. Even if they did not find all aspects of the care plan helpful, at least some part of the process was helpful in making them feel supported after leaving the hospital. 

#### 3.1.2. Illustrative Case

Leo (aged 43) went to the ED following an overdose. Leo described feeling listened to by the practitioner:

“I felt like he was really listening to me like sometimes health professional they’re writing stuff down and they’re not even making any eye contact to you. But when I spoke to him, he was thinking. You could tell he was thinking before he would answer, he wasn’t judging until he, until he understood what was going on and that felt like somebody was listening to you”.

Having taken the time to understand his needs, the practitioner then worked to mutually devise a care plan that included what would be helpful. Leo’s care plan included him being offered support from the crisis team and medication. The practitioner explored his previous issues with medication and made suggestions for medication that would not have the side effects that Leo had previously experienced. The practitioner took Leo’s previous experiences into account, validated his concerns about medication and came up with a jointly agreed upon plan. This was perceived positively by Leo, who described how this plan meant that things were “moving forward… things are happening… everything that was discussed is starting to happen”.

#### 3.1.3. Variation between Cases

Within this type, there was variation in whom was expected to initiate contact with the recommended services (i.e., patient/professional). The onus was typically on patients to take up the advice on offer, which they generally had done or intended to, as it was perceived as meaningful and appropriate advice. However, practitioners for a minority had arranged ongoing support on their behalf (with their GP or the university counselling service) which was perceived positively.

The majority of participants spoke positively about their care plan, citing at least one aspect of it that was helpful, almost exclusively focusing on professional sources of support. One participant differed from the rest of the participants in this type: Mollie (aged 17) described a thorough safety plan, which appeared more detailed than the care plans described by other participants in this type. Mollie described a safety plan that included professional sources of support, warning signs to help her notice future crises, distractions that she could use and what to do when in danger, covering the steps of evidence-based safety planning interventions. Mollie described this safety plan positively in contrast to previous care plans, as it was truly personalised and based on her experiences. The safety plan consisted of strategies that were tailored to her needs and experiences that Mollie could adapt over time. Mollie described how her safety plan was broken down into realistic steps. For instance, it specified how she could seek support from her family by communicating her distress to her mum with a specific emoji in a text message. Reaching out for help can be difficult, so this example shows how the practitioner had worked to break this down to a small and concrete step which felt achievable to Mollie. This appeared to offer a more in-depth care plan for Mollie, an outcome that may be due to her younger age or the hospital at which she presented adopting safety planning as an approach in psychosocial assessments.

### 3.2. Ideal-Type 2: Generic Care Plan 

Thirteen participants (41%) received a generic care plan in the ED.

#### 3.2.1. Ideal-Type Description

Thirteen participants reported feeling that their care plan was generic, typically providing recommendations to continue with strategies or services that they were already using (e.g., continue with therapy/contact care coordinator). They were critical, as these were things they had already known about or that a friend could have told them or were things that did not feel realistic for them to do. They had exhausted these options, leaving them feeling unsupported or that their distress had not been taken seriously. The care plan was perceived as a document that was given to them, rather than one that was made collaboratively, and was not considered helpful.

#### 3.2.2. Illustrative Case

Felicity (aged 39) went to the ED with suicidal ideation. When asked what was in her care plan, she stated that it told her to make an appointment with her GP to discuss her medication. She noted she had forgotten to do this. Secondly, it stated to continue using breathing exercises as a coping strategy, to which she commented, “well that ain’t working”, reflecting strategies that she had already tried and had not found helpful. Thirdly, it stated to seek support from others to attend appointments (e.g., Job Centre). To this, she responded, “well I’m not in any fit state to be doing that at the moment”. Felicity described feeling unable to focus on doing these things due to feeling too unwell. She described not wanting to burden others by asking for help. Overall, Felicity’s care plan provided suggestions where the onus was on her to carry them out, but the suggestions did not feel realistic. While these recommendations may, at face-value, appear reasonable, they were not perceived as helpful, as she did not feel able to take up these suggestions. The core difference between this and the previous type was that the recommendations felt ill-fitted to her current situation.

#### 3.2.3. Variation between Cases

Similar to Felicity, who felt unable to follow advice, two participants described how this would be easier if the practitioner were to arrange appointments (e.g., with GP) for them, rather than the onus being on them to do so, as it was often overwhelming to do so. Another participant said, “It is all very well making suggestions, but if those suggestions aren’t taken up then there isn’t anything that really came out of it”, emphasising the importance of feeling able to take up advice. 

In contrast to Felicity, five participants were already in contact with mental health services and described how their care plan advised them to contact services that they were already in touch with (e.g., contact care coordinator in office hours). Generic care plans often made people feel fobbed off and dismissed to other services. 

### 3.3. Ideal-Type 3: No Care Plan

Six participants (18%) reported not having received a care plan in the ED.

#### 3.3.1. Ideal-Type Description

Six participants reported they had not received a care plan from the ED. Some remembered being given crisis phone numbers but stated that this is not a care plan. They typically expressed how it would be useful to have a personalised plan in place to help support them after leaving the hospital, outlining the steps that they would take in the hours afterwards, including how to get home and what they would do, to help get them through the crisis.

#### 3.3.2. Illustrative Case

Lorna (64 years) went to the ED with suicidal ideation. She described never having received a care plan prior to leaving the hospital for suicidal thoughts. She stated that all she was offered was the crisis team phone number but did not perceive this to be a care plan. She suggested that she may not have appeared to be a serious danger to herself and speculated that this was the reason for not receiving a care plan. She reported that it would have been useful to have the details of whom to contact in a crisis and a plan for how to live in the days following the crisis.

#### 3.3.3. Variation between Cases

Similar to Lorna, two participants described being given a crisis number but emphasised that this is not a care plan: “That bunch of numbers that you give out to everybody is not going to save anyone”. One participant varied from the rest of cases in this type, as she did not remember whether she had received a care plan, while the rest explicitly stated not receiving one. Another did not receive a care plan due to leaving the ED before seeing liaison psychiatry.

### 3.4. Comparing Cases in the Ideal Types

Table 1 summarises the demographic data of participants in each type. This shows a higher average age for those who reported not receiving a care plan compared with other types. There was a greater proportion of men classified as receiving a personalised care plan compared with women, for whom a greater proportion were classified as receiving a generic care plan. A majority of participants classified as receiving a personalised care plan were not known to services, whereas those who were known to services tended to be classified as having received a generic care plan or no care plan at all. 

Table 2 compares the advice/information provided in care plans reported by participants in each type. Most advice referred to professional sources of support. The most frequent category reported by participants in all types was the provision of crisis numbers. Those with generic care plans more frequently reported advice to continue with existing care coordinator/therapy, compared with those with personalised care plans, who more frequently reported mental health service referrals or information about third-sector organisations that they were not aware of. 

### 3.5. Participant Recommendations for Care Plans

One important characteristic across all three types was that participants provided recommendations for optimising care plans. Participants expressed a need for holistic care planning that was not solely focused on professional support, indicating that they wanted it to include warnings signs for future crises, coping mechanisms, distractions and social support. Participants reported that care planning should include positive strategies to help improve the person’s life and creative options (e.g., social prescribing and voluntary organisations). Participants recommended that practitioners should obtain a holistic understanding of the person and explore what has or has not worked in the past to provide personalised recommendations.

## 4. Discussion

### 4.1. Main Findings

This study aimed to construct a typology of people’s perspectives on crisis care plans in EDs. This aim was achieved, and the results showed that people’s experiences of care plans fitted into three types following a self-harm/suicidal crisis. Personalised care plans provided at least one recommendation that the person considered helpful, typically something new to him/her, such as advice about a service/resource that he/she was not aware of, relevant to his/her difficulties. People intended to take up advice in personalised care plans, which often provided recommendations across multiple aspects of their life (e.g., GP for medication review, referral for therapy and employment/education support). Generic care plans provided advice or information that the person already knew about (e.g., Samaritans) or to continue using strategies or services that they had already tried (e.g., care coordinator) and were ill-fitted to their needs, making them less inclined to follow them. Six participants reported that they did not receive a care plan or were only provided with emergency phone numbers.

Personalised care plans often provided advice to try something new. Participants classified as receiving a personalised care plan tended not to be known to services, potentially making it easier for practitioners to provide recommendations that these people had not previously tried. In contrast, participants who received a generic care plan were typically known to services. This is perhaps unsurprising, as any advice or suggestions are more likely to have been provided previously, making it more challenging for practitioners to endorse services/resources that the person has previously tried and to identify alternative recommendations. Participants recommended that practitioners need to be more creative in making recommendations, such as social prescribing and voluntary sector organisations. This is one practical way in which practitioners may be able to make care plans more relevant for people, even those who have presented to ED previously and may have already exhausted the recommendations typically provided in care plans.

### 4.2. Findings in the Context of Other Studies

The findings are consistent with the research stating that collaboratively developed care plans are helpful [17,18,31,32]. The finding that participants who received generic care plans tended to be known to services is pertinent given the paucity of services for people who self-harm [4] and exclusionary practices, with patients often denied care based on diagnosis, risk and complexity [33]. When practitioners do not have suitable referrals/resources available to offer patients, this may lead to patients feeling dismissed and hopeless. This is concerning as leaving the hospital with some hope is valued by patients [16]. Care plans focused mostly on professional sources of support, such as referrals to mental health services. Participants endorsed a broader scope of care plans, which could include coping strategies; distractions; social support; and referrals, including to voluntary sector organisations, all of which could be recommended irrespective of limited National Health Service (NHS) resources for people who self-harm. Focusing on coping strategies, distractions and social support fit with safety planning interventions, which have been found to be effective in reducing suicidal behaviour [20,21,23].

One study found that 33% of patients reported not being provided with information about who to contact in a crisis, similar to the findings in this study [31]. Moreover, patients emphasised that crisis contact information was not enough to keep them safe. This reflects how care planning is often viewed by staff as a task to fulfil the organisational goal of producing a care plan, rather than it being a meaningful process from the patient’s perspective [16,34,35].

Participants described how practitioners should get to know the person and understand what has or has not worked previously in order to make them feel like the practitioner was listening and hopefully receive recommendations tailored to their experiences. This is consistent with the research exploring interactions between patients and practitioners in EDs which found that incorporating patients’ reasons for negative expectations towards treatment options (e.g., fear of talking about bereavement underlying decision not to accept therapy) led to greater patient acceptance of recommendations [19]. Participants often felt unable to take up advice, even when they agreed with it, as it could feel overwhelming, a finding that is in line with previous research [32]. In the present study, advice being broken down into small, manageable steps was perceived positively. For example, advice to spend time with friends/family could lead to fears of burdening loved ones. However, when the practitioner broke this down into specific steps (e.g., sending a text message to a named person), participants described greater willingness to try it. This is similar to safety planning, which identifies strategies that the person is willing to try [36] and barriers and solutions to help him/her follow his/her safety plan [20]. The findings demonstrate that care planning demands patient and professional buy-in and collaborative decision making [34].

### 4.3. Strengths and Limitations

The strengths of this study included a sample of people from urban (South East) and rural (South West) areas of England who had attended a number of different hospitals. The use of ideal-type analysis allowed us to provide illustrative cases to exemplify different experiences of care planning, which may be helpful for clinicians to consider how they can personalise patients’ care plans.

One limitation was the secondary analysis of data from studies where the main aims were not to explore care plans, so there was variation in the way participants were asked the interview questions. In the Relate study, participants were specifically asked about their views of their care plan, whereas, in the ASsuRED study, participants were asked more generally about their experiences of what was helpful and unhelpful in their experiences in the ED. The experiences of care plans came up spontaneously in the majority of the interviews and focus groups, so the extent to which this was discussed varied between participants. However, we excluded participants who did not discuss care plans in their interview/focus group. Some participants had recently been to the ED, whereas others had attended the ED several months or longer prior to the interview. Participants were those who agreed to be interviewed, so further research should be conducted with more heterogeneous samples. Data were susceptible to issues with participant recall, especially as participants were in crisis at the time when care planning was undertaken. The findings are based on what participants were aware of and could remember. We acknowledge that the sample was mostly female and White participants, so it is unknown how generalizable these findings are. Limited information was available on participants’ clinical characteristics. A further limitation is that the data were not available about the professional background of those who provided the care plans. Nevertheless, the findings provide important insight into people’s experiences of care plans and how they can be optimised.

### 4.4. Clinical Implications

Patients recommend a broader scope of care plans, which go beyond professional support, which is the focus of ED care plans currently. Recommendations fit with safety planning interventions, which have been identified as effective at reducing suicidal behaviour, through developing patients’ suicide-related coping skills and help seeking [21,23]. Care plans can include safety planning, incorporating warnings signs, coping mechanisms, distractions and social support, in addition to professional sources of support. Practitioners should be aware that many patients may not be experiencing current care plans as being helpful or relevant, thus possibly placing them at a greater risk of suicide. The key messages for practitioners to optimise care planning are as follows:Strive to develop holistic care plans in order to cover all aspects of the person’s life that they may need support with;Identify what has and has not worked previously for patients to ensure that advice is tailored to their experiences;Remember that advice is not always easy to follow when in crisis, so break down advice into small, achievable steps;Seek to explore positive strategies and solutions beyond NHS services (e.g., social prescribing).

## 5. Conclusions

Care planning is perceived as a supportive intervention by people presenting to EDs in a self-harm or suicidal crisis, provided it is personalised and provides recommendations that are perceived as relevant and realistic for the person to follow. Creating a meaningful care plan may be challenging for patients who are known to services or “frequent attenders” and are likely to have tried the options available to them. Patients who receive a generic care plan or who do not receive a care plan are unlikely to be getting the support they need, which may increase their suicide risk. Greater consistency in the provision of high-quality care plans for all patients is required. Further research is needed to explore how care planning can be optimized, including how practitioners can get to know the person holistically and provide personalised recommendations.

## Figures and Tables

**Table 1 ijerph-20-06883-t001:** A summary of demographic data for participants, organised by the ideal types.

	Ideal Type		
	Personalised Care Plan *n* = 13	Generic Care Plan *n* = 13	No Care Plan *n* = 6
	M (SD)	M (SD)	M (SD)
Age	30.43 (16.51)	29.60 (8.96)	42.89 (23.64)
	***n* (%)**	***n* (%)**	***n* (%)**
Gender			
Female	7 (54%)	11 (85%)	6 (100%)
Male	6 (46%)	2 (15%)	0 (0%)
Ethnicity			
White British	11 (84%)	10 (77%)	5 (83%)
Mixed	0 (0%)	1 (8%)	1 (17%)
Asian	1 (8%)	2 (15%)	0 (0%)
African	1 (8%)	0 (0%)	0 (0%)
Marital status			
Single	9 (69%)	9 (69%)	4 (66%)
Married/civil partnership	3 (23%)	1 (8%)	0 (0%)
Separated/divorced	1 (8%)	3 (23%)	1 (17%)
In a relationship	0 (0%)	0 (0%)	1 (17%)
Widowed	0 (0%)	0 (0%)	0 (0%)
Reason for Emergency Department (ED) presentation			
Self-harm	8 (62%)	7 (54%)	4 (67%)
Suicidal ideation	5 (38%)	6 (46%)	2 (33%)
Known to services			
Yes	4 (31%)	9 (69%)	6 (100%)
No	9 (69%)	4 (31%)	0 (0%)

M: Mean; SD: Standard Deviation.

**Table 2 ijerph-20-06883-t002:** Frequency of advice/information in care plans, as reported by participants in each of the types.

	Type		
	Personalised Care Plan *n* = 14	Generic Care Plan *n* = 14	No Care Plan *n* = 6
Professional support			
Crisis phone numbers (e.g., crisis team)	6	6	2
Advised about third-sector services (e.g., Listening Place)	2	-	-
Referral to mental health services, counselling or crisis team	5	-	-
Continue with care coordinator or therapy	1	5	-
Liaison psychiatry contacted General Practitioner (GP) on patient’s behalf	2	1	-
Advised patient to contact GP	2	2	-
Advised patient to attend Citizen’s Advice		1	-
Social support			
Advised to seek social support, e.g., phone friend/text mum	2	2	-
Employment/education			
Advised to seek support in work/university (e.g., Occupational Health)	1	1	-
Provided information about service for seeking employment	2	1	-
Provided information about courses/education	1	-	-
Distractions			
Advised to use distractions, e.g., breathing and harm-minimisation techniques	1	3	-
Miscellaneous			
Advised to reduce alcohol intake		1	-
Provided medication advice	3	1	-
Provided leaflets, e.g., online self-help resources	3	1	-

## Data Availability

The data are not publicly available due to ethical reasons.

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
