# Peer review of "An Ideal-Type Analysis of People’s Perspectives on Care Plans Received from the Emergency Department following a Self-Harm or Suicidal Crisis"

_ijerph, 2023, doi:10.3390/ijerph20196883_

Round 1

Reviewer 1 Report

Line 15 should explain the purpose of constructing a typology

Line 25: The authors’ findings are powerful in the sense that there could be increased suicide risk/attempts among those with generic or no care plan. As such, the authors should explain the implications of their findings after the last sentence currently written in the abstract.

Lines 31 and 32: The authors state that self-harm is the strongest risk factor for suicide. I think there needs to be exploration of the data found in who state

“The authors state that the We identified 40 risk factors that were examined in at least three independent samples (table 2). For significant associations, pooled ORs ranged from 2.2 to 4.0 in the sociodemographic domain. The strongest risk factors identified were social isolation (OR=4.0, 95% CI 2.1 to 7.7), unemployment (OR=3.8, 95% CI 2.7 to 5.2) and low socioeconomic status (OR=2.8, 95% CI 1.8 to 4.2). Risk factors within the family history domain were a family history of mental disorder (OR=5.2, 95% CI 1.9 to 14.1), suicide (OR=3.7, 95% CI 2.3 to 5.7) and attempted suicide (OR=2.8, 95% CI 1.5 to 5.0).

Within the clinical domain, a history of self-harm (OR=10.1, 95% CI 6.6 to 15.6) and a previous suicide attempt (OR=8.5, 95% CI 5.3 to 13.4; figure 1) were both strongly associated with suicide. We found strong associations for any mental disorder (OR=13.1, 95% CI 9.9 to 17.4) and any personality disorder (OR=6.8, 95% CI 4.7 to 9.8). By diagnosis, depression had the strongest association with suicide (OR=11.0, 95% CI 7.3 to 16.5), followed by borderline personality disorder (OR=9.0, 95% CI 5.6 to 14.4) and schizophrenia spectrum disorder (OR=7.8, 95% CI 4.5 to 13.5). Risk of suicide was comparable for alcohol use disorder (OR=3.2, 95% CI 2.3 to 4.4) and drug use disorder (OR=3.0, 95% CI 1.7 to 5.4).

For adverse life events, relationship conflict (OR=5.0, 95% CI 3.3 to 7.6), legal problems (OR=4.8, 95% CI 2.4 to 9.4) and family-related conflict (OR=4.5, 95% CI 2.0 to 10.3) had the strongest associations with suicide. By timing, adverse life events occurring within the previous month increased the risk of suicide 10-fold (OR=10.4, 95% CI 7.1 to 15.3).

(Favril et al., 2022, retrieved from https://www.ncbi.nlm.nih.gov/pmc/articles/PMC9685708/.

Reference

Favril L, Yu R, Uyar A, Sharpe M, Fazel S. Risk factors for suicide in adults: systematic review and meta-analysis of psychological autopsy studies. Evid Based Ment Health. 2022 Nov;25(4):148-155. doi: 10.1136/ebmental-2022-300549. Epub 2022 Sep 26. PMID: 36162975; PMCID: PMC9685708.

In the abstract the authors state their main goal is creating a typology of people’s perspectives of care plans. Subsequently, the goal is stated as “what contributes to care plans being experienced as helpful.” These need to align.

From the onset, the manuscript should make a clearer distinction between safety plan and care plan, as these terms are of central importance and therefore must be elaborated upon early in the document.

I indicated that I had concerns of ethical violations. It is of the utmost importance that the participants are debriefed and provided any and all guidance and resources needed after taking part in this type of study. Self-harm and suicide are sensitive issues and the research could have triggered the participants, leading to tragic endings. What was done to ensure the participants' safety? I would add what was done or would should have been done so future researchers know what is right when working with at-risk populations.

Author Response

Line 15 should explain the purpose of constructing a typology

Thank you for your comments. We have specified the purpose of constructing a typology (lines 13-14):

“We aimed to construct a typology of peoples’ perspectives of crisis care plans, to explore the range of experiences of care plans.”

Line 25: The authors’ findings are powerful in the sense that there could be increased suicide risk/attempts among those with generic or no care plan. As such, the authors should explain the implications of their findings after the last sentence currently written in the abstract.

We have added the implications to final sentence of abstract (lines 23-25):

“However, many people reported not receiving a helpful care plan, as it was ill-fitted to their needs or was not considered sufficient to keep them safe, which may mean these patients are at increased risk of repeat self-harm.“

Lines 31 and 32: The authors state that self-harm is the strongest risk factor for suicide. I think there needs to be exploration of the data found in who state

“The authors state that the We identified 40 risk factors that were examined in at least three independent samples (table 2). For significant associations, pooled ORs ranged from 2.2 to 4.0 in the sociodemographic domain. The strongest risk factors identified were social isolation (OR=4.0, 95% CI 2.1 to 7.7), unemployment (OR=3.8, 95% CI 2.7 to 5.2) and low socioeconomic status (OR=2.8, 95% CI 1.8 to 4.2). Risk factors within the family history domain were a family history of mental disorder (OR=5.2, 95% CI 1.9 to 14.1), suicide (OR=3.7, 95% CI 2.3 to 5.7) and attempted suicide (OR=2.8, 95% CI 1.5 to 5.0).

Within the clinical domain, a history of self-harm (OR=10.1, 95% CI 6.6 to 15.6) and a previous suicide attempt (OR=8.5, 95% CI 5.3 to 13.4; figure 1) were both strongly associated with suicide. We found strong associations for any mental disorder (OR=13.1, 95% CI 9.9 to 17.4) and any personality disorder (OR=6.8, 95% CI 4.7 to 9.8). By diagnosis, depression had the strongest association with suicide (OR=11.0, 95% CI 7.3 to 16.5), followed by borderline personality disorder (OR=9.0, 95% CI 5.6 to 14.4) and schizophrenia spectrum disorder (OR=7.8, 95% CI 4.5 to 13.5). Risk of suicide was comparable for alcohol use disorder (OR=3.2, 95% CI 2.3 to 4.4) and drug use disorder (OR=3.0, 95% CI 1.7 to 5.4).

For adverse life events, relationship conflict (OR=5.0, 95% CI 3.3 to 7.6), legal problems (OR=4.8, 95% CI 2.4 to 9.4) and family-related conflict (OR=4.5, 95% CI 2.0 to 10.3) had the strongest associations with suicide. By timing, adverse life events occurring within the previous month increased the risk of suicide 10-fold (OR=10.4, 95% CI 7.1 to 15.3).”

(Favril et al., 2022, retrieved from https://www.ncbi.nlm.nih.gov/pmc/articles/PMC9685708/.

We have now cited this important study (lines 30-33).

“There is an urgent need for interventions for people at increased risk of suicide, including those who self-harm, which is one of the leading clinical risk factors for suicide, associated with a 10-fold increase in the odds of suicide [2].”

In the abstract the authors state their main goal is creating a typology of people’s perspectives of care plans. Subsequently, the goal is stated as “what contributes to care plans being experienced as helpful.” These need to align.

Thank you for flagging the inconsistency here. We have corrected this in the introduction to state that (lines 81-82):

“The aim of the present study was to construct a typology of people’s perspectives of crisis care plan.”

From the onset, the manuscript should make a clearer distinction between safety plan and care plan, as these terms are of central importance and therefore must be elaborated upon early in the document.

We have added (lines 66-70):

“While care plans outline the support and interventions available for the person, safety plans tend to break this down further, specifying a hierarchical series of steps that can be followed in times of crisis, including non-professional coping strategies (e.g. things person can do in their environment or people in their social network they can contact for support).”

I indicated that I had concerns of ethical violations. It is of the utmost importance that the participants are debriefed and provided any and all guidance and resources needed after taking part in this type of study. Self-harm and suicide are sensitive issues and the research could have triggered the participants, leading to tragic endings. What was done to ensure the participants' safety? I would add what was done or would should have been done so future researchers know what is right when working with at-risk populations.

Thank you for raising these important points. In the ‘Ethical considerations’ section we have expressed how these were managed during data collection (lines 148-152):

“Participants were de-briefed at the end of their interview or focus group to ensure their participation had not left them feeling distressed. A risk protocol was in place for researchers to follow in the event that they had concerns about the safety of a participant, to ensure that appropriate support was in place (e.g. via signposting or contacting a care provider).”

Reviewer 2 Report

TOPIC:

An ideal-type analysis of people’s perspectives of care plans re- 2

ceived from the Emergency Department following a self-harm 3

or suicidal crisis

Thank you for inviting me to review this article. My comments are as follows:

TITLE

The title is satisfactory and represents the whole article

ABSTRACT

It is better to write complete sentences without listing- Background:, Methods:, Results:, and Conclusions:

INTRODUCTION

The introduction is satisfactory

MATERIALS AND METHODS

1.      Sample is satisfactory

2.      The process of data analysis is clearly stated

RESULTS

 Interpretation of the results is fascinating.

DISCUSSION

The discussion was fruitful.

CONCLUSION

 It is suggested that the conclusion be written in one paragraph only

Author Response

TOPIC: An ideal-type analysis of people’s perspectives of care plans received from the Emergency Department following a self-harm or suicidal crisis

Thank you for inviting me to review this article. My comments are as follows:

TITLE

The title is satisfactory and represents the whole article

We thank the reviewer for their comments.

ABSTRACT

It is better to write complete sentences without listing- Background:, Methods:, Results:, and Conclusions:

We have removed the headings from the abstract.

INTRODUCTION

The introduction is satisfactory

MATERIALS AND METHODS

  1. Sample is satisfactory
  2. The process of data analysis is clearly stated

RESULTS

 Interpretation of the results is fascinating.

DISCUSSION

The discussion was fruitful.

Thank you for your comments.

CONCLUSION

It is suggested that the conclusion be written in one paragraph only

We have revised the conclusion so that it is now in one paragraph only. Prior to this, we have also added a ‘Clinical Implications’ section to summarise key learning points for practitioners.

Reviewer 3 Report

This is a manuscript about the experience of care plans after being treated in an emergency department.

The abstract and introduction are adequate for the study.

Regarding the methodology. There are some aspects to consider. They included 4 participants who were not directly asked about the received care plans and were drawn from focus groups. Considering there are only 4 of the 32 participants, I recommend excluding them and leaving only those participants who were interviewed and questioned specifically on the topic of interest. It is necessary to include the questions asked by the interviewers. It is necessary to clarify whether there was any relationship between the interviewees and the hospitals in which the participants were treated. It is necessary to include the reason for admission of the participants (self-harm, suicide ideation, suicide attempt) as well as, other clinically relevant information, for example, occasions in which the person went to the ED for the same reason, if they were in mental health care and so on. It is necessary to mention the clinical characteristics of the participants briefly. All the above can qualify the perception of the usefulness of care plans in the ED.

Regarding the results. It deals with a mostly female and white sample, which must be considered as a limitation.

The age range of the participants was 18-76. But Mollie's case is 17 years old, which is contradictory to what was reported.

On the qualitative results, there are a lot of inconsistencies. Regarding type 1, it is mentioned that the participants considered that they were provided with a new resource, while in table 2 it only says that it was 2 participants who reported it. It is mentioned that the plans typically focused on aspects of mental health (it is necessary to quantify them), and later it mentions that “often spanned across multiple domains” without quantifying them. He mentions that the recommendations were relevant, without quantifying them. Lastly, it mentions that some did not find the care plan helpful, which is contradictory to the main finding that mentions just the opposite. It seems that what makes the difference is the type of dialogue or psychosocial intervention received in the ED.

In this same regard, in a single case “Mollie” received a care plan divided into stages, including multiple aspects of her life, not just mental health. It is a relevant finding, but it is a single case. It is necessary to consider Mollie's characteristics that could have made her more receptive to information (adolescent). On the other hand, this point was included in the conclusions, which I consider inappropriate as it is a single case of the type 1 group.

Regarding the discussion. There is no mention of a relevant aspect that could make the findings understood in another way. Most of the participants in the type 1 group were unaware of services or resources for their care. In contrast, the other two groups already knew about the resources. This data could be associated with the times in which people have previously gone to the ED and the time they have been receiving mental health care. It is necessary to consider this aspect.

It is mentioned again that a finding was that the plans of the type 1 group included different aspects of their lives, not only mental health, but this was not quantified, and when it was mentioned that the care plans focused on aspects of mental health.

Discussing the finding of Mollie's case is not relevant, since it only occurred in one participant. It leaves other relevant aspects such as those mentioned in previous lines without discussing.

The conclusions are based on aspects not quantified in the study, which are possibly derived from a single case with particular characteristics such as Mollie's case.

Author Response

This is a manuscript about the experience of care plans after being treated in an emergency department.

The abstract and introduction are adequate for the study.

We thank the reviewer for their comments.

Regarding the methodology. There are some aspects to consider. They included 4 participants who were not directly asked about the received care plans and were drawn from focus groups. Considering there are only 4 of the 32 participants, I recommend excluding them and leaving only those participants who were interviewed and questioned specifically on the topic of interest. It is necessary to include the questions asked by the interviewers.

We confirm that all participants included in the analysis did explicitly discuss their care plan. We state in the Data Collection section (lines 114-115):

“Participants were excluded from the present study where they didn’t discuss a care plan”.

It is necessary to clarify whether there was any relationship between the interviewees and the hospitals in which the participants were treated.

We have clarified that the interviewers were independent from the hospital in the data collection section (lines 105-107):

“Participants took part in a semi-structured interview/focus group to explore their experiences of treatment following a self-harm/suicidal crisis in EDs, with a researcher who was independent of the hospital in which participants were treated.”

It is necessary to include the reason for admission of the participants (self-harm, suicide ideation, suicide attempt) as well as, other clinically relevant information, for example, occasions in which the person went to the ED for the same reason, if they were in mental health care and so on. It is necessary to mention the clinical characteristics of the participants briefly. All the above can qualify the perception of the usefulness of care plans in the ED.

Thank you for raising this important point. Although we have limited information clinical information available for the sample, we have added the reason for ED presentation (self-harm / suicidal ideation). This is now presented in Table 1, and we state for each optimal case the reason for their ED presentation. We also acknowledge the limited clinical information as a limitation (line 384-385):

“Limited information was available on participants’ clinical characteristics.”

Regarding the results. It deals with a mostly female and white sample, which must be considered as a limitation.

Thank you for raising this important point. We have now acknowledged this in the limitations (lines 383-385):

“We acknowledge that the sample was mostly female and White participants so further research should be conducted with more heterogeneous samples.”

The age range of the participants was 18-76. But Mollie's case is 17 years old, which is contradictory to what was reported.

Thank you for bringing this typo to our attention. This has been corrected so the age range has been corrected to 17-76 (line 154).

On the qualitative results, there are a lot of inconsistencies. Regarding type 1, it is mentioned that the participants considered that they were provided with a new resource, while in table 2 it only says that it was 2 participants who reported it.

The new resources refer to numerous domains in Table 2. We realise this was misleading as one of them explicitly referred to the person being unaware of the service. Therefore, we have amended the category ‘Advised about third sector services that they weren’t aware of’ to ‘Advised about third sector services’. We don’t know in all cases which services participants were aware of, so we do not quantify this, but it was a common characteristic in this type that participants referred to recommendations that were new to them.

It is mentioned that the plans typically focused on aspects of mental health (it is necessary to quantify them), and later it mentions that “often spanned across multiple domains” without quantifying them. He mentions that the recommendations were relevant, without quantifying them.

We have elaborated on the domains that recommendations covered (lines 167-169):

“Recommendations often spanned across multiple types of services (e.g., GP, talking therapies, crisis lines), and having numerous recommendations may have increased the likelihood of some of these being perceived as useful. Moreover, recommendations went beyond mental health support (e.g. employment or education advice) making a holistic plan covering a range of areas of the person’s life. “

Lastly, it mentions that some did not find the care plan helpful, which is contradictory to the main finding that mentions just the opposite. It seems that what makes the difference is the type of dialogue or psychosocial intervention received in the ED.

To clarify, the point we are making is that participants with a personalised care plan found at least one thing helpful from their care plan. We consider it important to acknowledge that not all participants found all parts of their care plan helpful, but there was something in it that was meaningful for them. Therefore they did overall find their care plan helpful, even if there were components of it that they did not find helpful. This is stated in the final sentence of the ideal-type description (lines 173-174):

“Even if they didn’t find all aspects of the care plan helpful, at least some part of the process was helpful in making them feel supported after leaving hospital.”

In this same regard, in a single case “Mollie” received a care plan divided into stages, including multiple aspects of her life, not just mental health. It is a relevant finding, but it is a single case. It is necessary to consider Mollie's characteristics that could have made her more receptive to information (adolescent). On the other hand, this point was included in the conclusions, which I consider inappropriate as it is a single case of the type 1 group.

In the results, we have added in possible explanations for this variation (lines 211-213):

“This appeared to offer a more in depth care plan for Mollie, which may be due to her younger age or the hospital she presented adopting safety planning as an approach in psychosocial assessments.”

We think it is important to reflect on our learning from Mollie’s case in this discussion. A key aspect of ideal-type analysis includes comparison of cases – to look for similarities and differences between cases - to ensure the analysis captures the inevitable variation within ideal types (Stapley, O’Keeffe et al., 2021). Although this is one case it is important to capture this important variation which provides valuable insight into potential ways in which care plans can be personalised. Moreover, the findings from Mollie’s case are further supported by the findings in ‘Participant Recommendations for Care Plans’. This section summarises participant recommendations for how care plans could be improved, which although the other participants didn’t experience, they suggested that many of the components that Mollie described would have been potentially helpful for them (e.g. focus on coping strategies, social support and more creativity around recommendations).

We have revised the conclusion so it is less focused on the findings specific to Mollie and draws conclusions more generally from all of the types identified in this study (lines 408-417).

“Care planning is perceived as a supportive intervention by people presenting to EDs in a self-harm or suicidal crisis, provided it is personalised and provides recommendations perceived as relevant and realistic for the person to follow. Creating a meaningful care plan may be challenging for patients who are known to services or ‘frequent attenders’, who are likely to have tried the options available to them. Patients who receive a generic care plan or who don’t receive a care plan are unlikely to be getting the support they need, which may increase their suicide risk. Greater consistency in the provision of high quality care plans for all patients is required. Further research is needed to explore how care planning can be optimized, including how practitioners can get to know the person holistically and provide personalised recommendations.”

Regarding the discussion. There is no mention of a relevant aspect that could make the findings understood in another way. Most of the participants in the type 1 group were unaware of services or resources for their care. In contrast, the other two groups already knew about the resources. This data could be associated with the times in which people have previously gone to the ED and the time they have been receiving mental health care. It is necessary to consider this aspect.

The second paragraph of the discussion considers the important point that type 2 were typically known to services, potentially making it more difficult for practitioners to offer new/meaningful recommendations, compared with those who were not known to services. We have elaborated on how practitioners may be able to respond in such cases where a person is known to services and has received previous care plans (lines 322-326):

“Participants recommended that practitioners need to be more creative in recommendations, such as social prescribing and voluntary sector organisations. This is one practical way in which practitioners may be able to make care plans more relevant for people, even those who have presented to ED previously and may have already exhausted the recommendations typically provided on care plans.”

It is mentioned again that a finding was that the plans of the type 1 group included different aspects of their lives, not only mental health, but this was not quantified, and when it was mentioned that the care plans focused on aspects of mental health.

We have further elaborated on the different types of advice in care plans in the ideal type description (lines 167-170):

“Recommendations often spanned across multiple types of services (e.g., GP, IAPT, crisis lines), and having numerous recommendations may have increased the likelihood of some of these being perceived as useful. Moreover, recommendations went beyond mental health support (e.g. employment or education advice) making a holistic plan covering a range of areas of the person’s life.

Discussing the finding of Mollie's case is not relevant, since it only occurred in one participant. It leaves other relevant aspects such as those mentioned in previous lines without discussing.

As noted above, we think discussion of Mollie’s case is relevant, especially as she is a case with one of the most positive experiences of her care plan, which means we are keen to emphasise the learning when care planning has gone well. Her experiences are also backed up by the recommendations for care planning from participants in the whole sample. The final stage of the data analysis (Comparing similarities and differences in cases within and between the ideal-types) enabled us to explore commonalities between the perspectives of participants in all three types, which included the recommendations they provided for how care plans could be improved. These have been summarised in Section 3.5 (Participants recommendations for Care plans) (lines 293-300).

3.5. Participant Recommendations for Care Plans

One important characteristic across all three types was that participants provided recommendations for optimising care plans. Participants expressed a need for holistic care planning, not solely focused on professional support, to include warnings signs for future crises, coping mechanisms, distractions and social support. Participants reported that care planning should include positive strategies to help improve the person’s life and creative options (e.g., social prescribing and voluntary organisations). Participants recommended practitioners should get a holistic understanding of the person and explore what has/has not worked in the past, to provide personalised recommendations.

The conclusions are based on aspects not quantified in the study, which are possibly derived from a single case with particular characteristics such as Mollie's case.

We have added a Clinical Implications section, which now summarises key learning for practitioners, including some of the key points of what we can learn from Mollie’s case (as well as the Participant Recommendations for Care Plans, as noted above). The conclusion has been revised substantially to focus on the key points from the study, so that Mollie’s case is not the focus of the conclusion (lines 407-417):

“Care planning is perceived as a supportive intervention by people presenting to EDs in a self-harm or suicidal crisis, provided it is personalised and provides recommendations perceived as relevant and realistic for the person to follow. Creating a meaningful care plan may be challenging for patients who are known to services or ‘frequent attenders’, who are likely to have tried the options available to them. Patients who receive a generic care plan or who don’t receive a care plan are unlikely to be getting the support they need, which may increase their suicide risk. Greater consistency in the provision of high quality care plans for all patients is required. Further research is needed to explore how care planning can be optimized, including how practitioners can get to know the person holistically and provide personalised recommendations.”